# Effects of Transient Administration of the NMDA Receptor Antagonist MK-801 in *Drosophila melanogaster* Activity, Sleep, and Negative Geotaxis

**DOI:** 10.3390/biomedicines11010192

**Published:** 2023-01-12

**Authors:** Thiago C. Moulin, Tijana Stojanovic, Rasika P. Rajesh, Tirusha Pareek, Laura Donzelli, Michael J. Williams, Helgi B. Schiöth

**Affiliations:** 1Department of Surgical Sciences, Division of Functional Pharmacology and Neuroscience, Uppsala University, 751 24 Uppsala, Sweden; 2Department of Experimental Medical Science, Faculty of Medicine, Lund University, 221 84 Lund, Sweden

**Keywords:** fruit fly, invertebrates, glutamate receptor, dizocilpine, circadian activity, climbing behavior, psychiatric models, translational models

## Abstract

MK-801, also called dizocilpine, is an N-methyl-D-aspartate (NMDA) receptor antagonist widely used in animal research to model schizophrenia-like phenotypes. Although its effects in rodents are well characterised, little is known about the outcomes of this drug in other organisms. In this study, we characterise the effects of MK-801 on the locomotion, sleep, and negative geotaxis of the fruit fly *Drosophila melanogaster*. We observed that acute (24 h) and chronic (7 days) administration of MK-801 enhanced negative geotaxis activity in the forced climbing assay for all tested concentrations (0.15 mM, 0.3 mM, and 0.6 mM). Moreover, acute administration, but not chronic, increased the flies’ locomotion in a dose-dependent matter. Finally, average sleep duration was not affected by any concentration or administration protocol. Our results indicate that acute MK-801 could be used to model hyperactivity phenotypes in *Drosophila melanogaster*. Overall, this study provides further evidence that the NMDA receptor system is functionally conserved in flies, suggesting the usefulness of this model to investigate several phenotypes as a complement and replacement of the rodent models within drug discovery.

## 1. Introduction

*Drosophila melanogaster* has become a popular animal model for neuroscience, as there are a number of parallels in neurobiological processes, including cell membrane excitability, regulations of neuronal functions, and shared classes of neurotransmitters [1,2]. Moreover, there are remarkable genetic similarities, as homologues of 77% of human disease-associated genes can be found in *Drosophila* [3]. Due to its versatility and simpler brain, which contains only 100 thousand neurons in comparison to the 86 billion neurons in humans and 70 million in mice, the fly is currently a consolidated reductionist model for brain pathologies [4].

Accordingly, an increasing number of studies have examined the effects of interventions during behavioural assays [5]. Flies have also increasingly been used as models for mental disorders, such as depression [6,7], autism [8,9], and alcohol addiction [10,11], among others [4]. Moreover, due to the conserved structural organisation of sensorimotor control [12], the fruit fly may also be a suitable organism for pharmacological models of schizophrenia-like phenotypes.

The functionality of NMDA receptors from *Drosophila* is also similar to rodents, as the invertebrates possess highly conserved homologues of the main mammalian genes encoding the NMDA receptor subunits [13,14,15]. In mammals, three families of NMDA receptor subunits are known, namely NR1, NR2, and NR3. The NR1 subunit is a fundamental component of NMDA receptors, and its expression can be found throughout the nervous system [16]. The NR2 subunit is particularly important, as it is responsible for regulating the pharmacological and biophysical properties of the channel, such as its kinetics, modulation by glycine, Mg2+ inhibition, and glutamate affinity [17,18]. NR3 subunits are able to assemble with NR1–NR2 complexes to modulate channel function; however, they are not part of most endogenous NMDA receptors [18]. In flies, homologues for the NR1 and NR2 subunits have been described, which seem to share important genetic and structural features [19]. For instance, the NR1 domain for glycine binding, as well as the NR2 domain for glutamate binding, have well-conserved amino acid sequences [15,20,21]. Comparable functional properties of the NMDA receptors in signalling pathways and neuronal plasticity have also been comprehensively described [21,22,23].

MK-801, or dizocilpine, is one of the most common NMDA receptor antagonists employed in pre-clinical studies. Despite being a neuroprotectant in trauma, stroke and Parkinsonism models, it can induce psychotic behaviour. Exposure to this antagonist also causes neuronal degeneration in microglia, axon terminals, entorhinal cortices, the retrosplenial cortex, and neurons in the pyriform in rats [24,25]. It is often used to model phenotypes of schizophrenia in animals, such as sensorimotor gating deficits [26], decreased pre-pulse inhibition [27], and impairment of social behaviour [28]. Moreover, exposure to MK-801 has been found to have anxiolytic effects [29] as well as locomotor stimulant effects [30,31]. Chronic administration of MK-801 can also lead to the decreased brain expression of NMDA receptors in rodents, resulting in biochemical alterations in the brain comparable to those encountered in schizophrenia [32].

Likewise, there are many pharmacological similarities between the interaction of MK-801 and NMDA receptors for both mammalian and invertebrate organisms. For example, it has been demonstrated that NMDA-dependent processes in invertebrates can be blocked by MK-801 administration [23,33,34,35]. Additionally, like the mammalian counterpart, MK-801 targets the same asparagine residue of the NR1 subunit, a binding site conserved in all studied invertebrate NR1 homologues [20,21,22,35].

The choice of NMDA receptor antagonists, such as ketamine, PCP, and MK-801, have also been based on the NMDA hypofunction theory, which suggests that NMDA receptors, particularly those of interneurons, are partly absent in schizophrenic individuals [36]. In principle, the loss of inhibition from interneurons leads to the overstimulation of glutamatergic and monoaminergic neurotransmitter systems, inducing a range of functional consequences [37]. Thus, due to the parallels of the dopaminergic and serotonergic behavioural function [2], as well as the conserved structural organisation of sensorimotor control [12], we evaluated if the fruit fly would also be a suitable organism for such a pharmacological model.

Previous reports testing the effects of the continuous administration of MK-801 to *Drosophila melanogaster* have shown that it can induce sleep impairment and increased levels of activity [38]. However, widespread approaches in other pre-clinical models usually consist of transient pharmacological intervention, where NMDA antagonists are withdrawn prior to behavioural testing. Thus, in this study, we administered the NMDA antagonist MK-801 acutely (24 h) and chronically (7 days) prior to evaluating its effects on general activity, sleep characteristics, and forced climbing locomotion (negative geotaxis). These assays were chosen to investigate features that are essential for rodent experiments, such as alterations in locomotion due to stress and general activity changes [39]. To the best of our knowledge, this is the first report assessing the effects of transient MK-801 on locomotor-based assays in *Drosophila melanogaster*.

## 2. Materials and Methods

### 2.1. Fly Strains and Maintenance

For the experiments, 5 to 7-days-old CSORC-strain *Drosophila melanogaster* flies were used (originated from CantonS and OregonR-C flies, Bloomington Stock Center, Bloomington, IN, USA) and kept at 25 °C with 12:12 h light/dark and 60% humidity. The flies were fed with the Fisherbrand Jazz-Mix *Drosophila* food, complemented with 8.3% yeast extract (both from Fisher Scientific, Gothenburg, Sweden).

### 2.2. Pharmacology

Flies were treated with the NMDA antagonist MK-801/Dizocilpine (Sigma-Aldrich, Stockholm, Sweden). Firstly, forced climbing and general activity assays were performed by previously treating the flies with MK-801 acutely (24 h) or chronically (7 days) at 0.6, 0.3, or 0.15 mM concentrations homogenised in the fly food. Drug concentrations were chosen based on a previous publication [38], where the continuous application of ~0.3 mM MK-801 was sufficient to induce sleep deficits in flies, and 0.15 mM and 0.6 mM concentrations were added to observe possible dose–response effects. For all experiments, control groups were made by adding an equivalent amount of vehicle solution (distilled water) to the fly food.

### 2.3. General Activity and Forced Climbing Assays

For the general activity experiments in *Drosophila melanogaster*, we used the Drosophila Activity Monitor System (DAMS, TriKinetics Inc., Waltham, MA, USA) as previously described [40]. Briefly, after treating the flies with MK-801 acutely (24 h) or chronically (7 days), flies were anaesthetised with CO_2_ and put individually in plastic tubes, which were then placed horizontally in a detection system. The tube ends were sealed with regular fly food and a small cotton bud. Flies were maintained in a 12:12 h light/dark cycle for 3 days, starting at 8 a.m. Next, the raw data were converted to CSV files using DamFileScan software. The Sleep and Circadian Analysis MATLAB Program (SCAMP) by Vecsey Lab was used to analyse the data and evaluate different aspects of activity behaviour.

As described in previous reports [6,41], the forced climbing assay was performed using the MB5 MultiBeam Activity Monitor coupled with the Vortexer Mounting Plate, both from TriKinetics. After being anaesthetised, flies were put individually in glass tubes sealed with pieces of cotton and a lid. The tubes were then put vertically in the device, stimulating the flies to climb upwards due to their natural negative geotaxis behaviour. The device was programmed to shake the flies down every minute for 4 s, during a five-hour-long session, performed from 1 p.m. to 6 p.m. For analysis, DamFileScan was used, and the generated CSV files were further evaluated in Microsoft Excel. The beam-to-beam transitions (named ‘moves’) during the shaking period were excluded, and the number of beam crosses per second was averaged for each hour to estimate the overall climbing activity.

### 2.4. Statistical Analyses

In experiments testing the effect of treatment groups over time in relation to the controls (Figure 1B,C and Figure 2A,C), a two-way ANOVA analysis was applied, followed by Holm–Sidak post hoc tests. A one-way ANOVA was employed to compare averages from multiple treatment groups (Figure 2B,D and Figure 3), and dose–response effects were analysed using a linear trend post hoc test. All analyses were two-sided, and the results were considered statically significant if *p* < 0.05. The statistical tests used for each experiment are also described in the legends of their respective figures. All analyses were performed using GraphPad Prism 8 software. All data met the assumptions of the applied tests (i.e., independence of cases, normal distribution, and homogeneity of variance—the latter two checked using default analyses from Prism 8 software).

## 3. Results

### 3.1. Forced Climbing Assay

We administered three concentrations of MK-801 (0.15, 0.30, or 0.60 mM) via the food of wild-type adult flies, both acutely (24 h) and chronically (7 days), followed by a forced climbing assay (Figure 1A). All tested concentrations significantly increased locomotion during the forced climbing assay, where the flies were mechanically tapped down every minute for five hours, and the average movement due to innate negative geotaxis was measured (Figure 1B,C) (two-way ANOVA, *p* < 0.0001 and *p* = 0.0002 for 24 h and 7 dtreatment effects, respectively, confirmed by Holm–Sidak post hoc tests for each group; results shown in the figure). No effects of time or drug–time interactions were observed, suggesting that flies were not fatigued in our protocol, as there was no significant change in moves over time, and that the treatment did not differentially induce changes over time (p_time_ = 0.946 and 0.865; p_interaction_ = 0.998 and 0.999; for 24 h and 7 d treated flies, respectively).

### 3.2. General Activity

Upon finishing either acute or chronic treatments, general locomotion analysis was performed using the Drosophila Activity Monitoring System (DAMS) over three days (Figure 2). Feeding flies with MK-801 for 24 h significantly increased general locomotion for all tested concentrations, most notably during the initial times of the light period (Figure 2A). Evaluating the average daily activity revealed that such increases in activity levels were induced in a dose-dependent manner (Figure 2B).

Interestingly, 7-day MK-801 treatment induced a small but significant reduction in the general locomotion compared to controls, especially between ZT 6 and 10 time points (Figure 2C). However, this reduction was insufficient to produce significant differences in the average daily activity nor to exhibit any dose–response dependencies (Figure 2D).

### 3.3. Sleep Duration

Next, we analysed sleep activity after acute or chronic transient administration of MK-801. Interestingly, neither intervention significantly altered daily sleep time (Figure 3), differing from previous reports employing continuous MK-801 administration [38].

## 4. Discussion

In this study, we implemented an NMDA inhibition model for sensorimotor deficits by administering the MK-801 compound [27]. The NMDA inhibition model has been widely used in rodent research, but its effects in adult *Drosophila melanogaster* have not been characterised, apart from one study demonstrating that the continuous administration of MK-801 in the flies’ food reduces sleep duration [38]. Here, we administered MK-801 for 24 h or 7 days, which was withdrawn prior to the tests evaluating forced climbing locomotion (negative geotaxis), general activity, and sleep characteristics. The concentrations (0.15, 0.3, and 0.6 mM) were chosen based on a previous report employing MK-801 in *Drosophila* (~0.3 mM) [38].

We demonstrated that both acute and chronic treatments in all tested concentrations induced increased locomotion in the forced climbing assay. Interestingly, when assessing the overall activity for three days after drug administration, only the acute treatment was able to increase locomotion in a dose–response manner. This difference may be due to NMDA receptor homeostatic adjustments induced by chronic changes in neuronal excitability [42,43], which might have induced behavioural adaptations after the long-term MK-801 treatment. Moreover, although differences could be observed at specific points throughout the day, neither protocol of NMDA inhibition was able to interfere with the overall sleep time significantly.

It is worth noting that our findings differ from a previous study that reported reduced sleep time in flies treated with similar doses of MK-801 [38]. We attribute this discrepancy to the different administration protocols used in the two studies, as we removed the treatment before the 3-day assessment of general activity and sleep in our study. This interpretation is supported by data from rodents, which have shown that NMDA antagonists consistently impair electrophysiological sleep features [44,45] but are unable to reduce the sleep–wake times in several instances [46,47]. Therefore, we believe that the transient nature of our drug administration protocol likely hindered the influence of MK-801 on *Drosophila* sleep time, although it is possible that other sleep features not measured in this study were impaired.

Additional studies are needed to fully characterise the extent of similarities between fly and rodent behavioural impairments due to the administration of MK-801. However, our results indicate that the marked characteristics of rodent models employing this drug, such as alterations in locomotion due to stress and general activity [39], can be reproduced in *Drosophila melanogaster*. Taken together, these data suggest that the NMDA receptor’s influence on behaviour is functionally conserved in the fly, which provides further evidence that this biological system is highly conserved through evolution.

## 5. Conclusions

Overall, our findings suggest that *Drosophila melanogaster* may be a valuable tool for studying schizophrenia-like locomotor phenotypes induced by the acute administration of MK-801. Moreover, due to the benefits of the use of *Drosophila* as a model organism, including the simplicity of the experimental setup and the ability to perform large-scale genetic manipulations, we believe it could be an attractive model for studying the effects of NMDA antagonists on behaviour and for identifying potential therapeutic targets for schizophrenia.

## Figures and Tables

**Figure 1 biomedicines-11-00192-f001:**
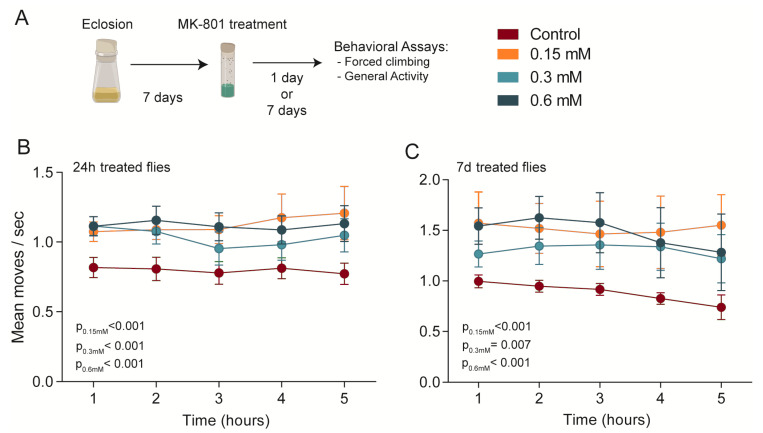
NMDA inhibition effects on *Drosophila melanogaster* climbing behaviour. (**A**) Protocol for acute (24 h) or chronic (7 days) MK-801 pharmacological treatment and subsequent behavioural testing. (**B**) 24 h treated and (**C**) 7 d treated flies showed increased locomotion in the forced climbing test. Two-way ANOVA analysis was used for forced climbing (24 h intake: n_[all groups]_ = 16; 7 d intake n_[all groups]_ = 8). Values are shown as mean ± SEM.

**Figure 2 biomedicines-11-00192-f002:**
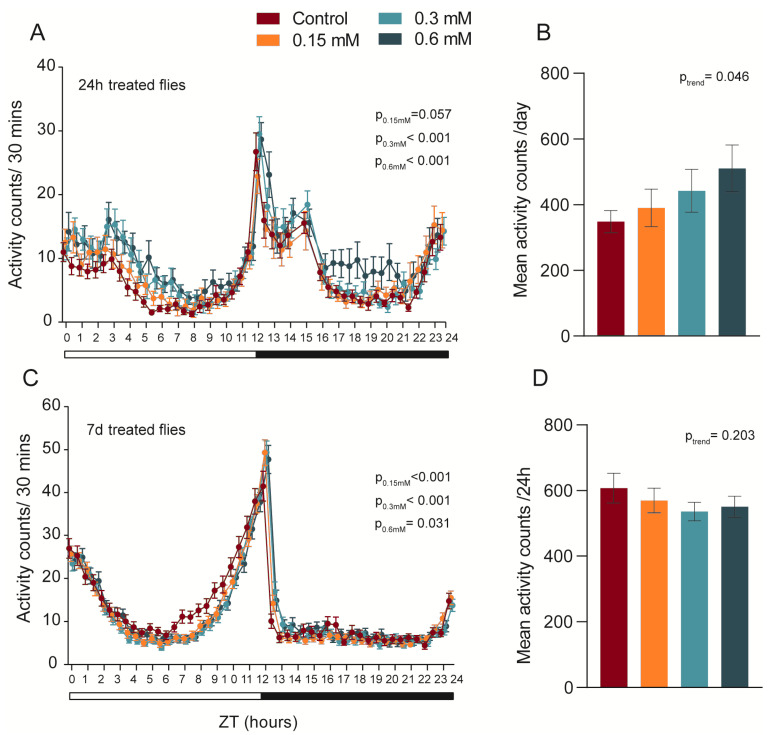
NMDA inhibition effects on general activity and circadian rhythm. (**A**) Flies treated acutely demonstrated an increase in general activity, mostly during the light period. (**B**) These effects were significantly dose-dependent. (**C**) Chronic MK-801 intake, on the other hand, induced a significant reduction in general locomotion, mainly during the last hours of the light phase. (**D**) However, it was not enough to present differences in mean daily activity (right panel). For daily activity counts analysis and dose–response average activity analysis (24 h intake: n_[control/0.6mM]_ = 26, n_[0.15mM]_ = 25, n_[0.3mM]_ = 29; 7 d intake: n_[control]_ = 76 n_[0.15mM]_ = 65, n_[0.3mM]_ = 87, n_[0.6mM]_ = 82), two-way ANOVA and one-way ANOVA with linear trend post hoc were applied, respectively. Values are shown as mean ± SEM.

**Figure 3 biomedicines-11-00192-f003:**
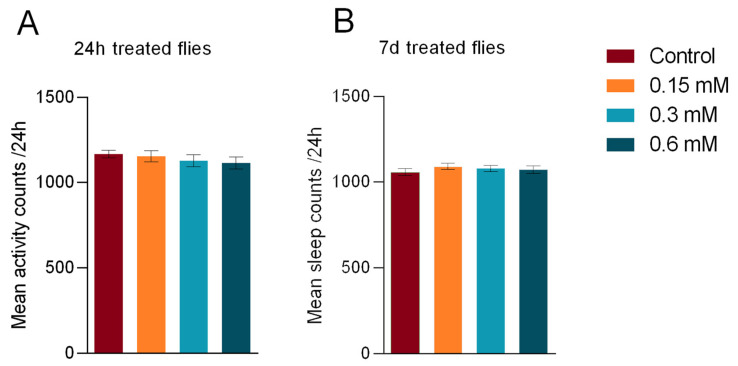
NMDA-inhibition effects on daily sleep time. (**A**) Despite inducing increased activity, 24 h MK-801 treatment was unable to evoke significant changes in average daily sleep counts, independent of concentration. (**B**) Similarly, 7 d intake of MK-801 at the tested concentrations did not alter sleep. Sample sizes are the same as those described in Figure 2. One-way ANOVA with linear trend post hoc was applied for statistical calculations, resulting in *p* > 0.05 for all comparisons. Values are shown as mean ± SEM.

## Data Availability

The data presented in this study are available in the Appendix A containing GraphPad Prism files.

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
