# Peer review of "Effects of Transient Administration of the NMDA Receptor Antagonist MK-801 in Drosophila melanogaster Activity, Sleep, and Negative Geotaxis"

_biomedicines, 2023, doi:10.3390/biomedicines11010192_

Round 1

Reviewer 1 Report

The manuscript entitled Effects of transient administration of the NMDA receptor antagonist MK-801 in Drosophila melanogaster activity, sleep, and negative geotaxis is presented for the peer review.  In  the study,authors analyzed the effects of MK-801 on the locomotion, sleep, and negative geotaxis of the fruit fly Drosophila melanogaster. Authors used fruit fly as the popular animal model for neuronal studies. You did great job to perform such interesting project.

Good idea to use combined effects of MK-801 on general activity, sleep characteristics, and fnegative geotaxis. 

I have several suggestions to authors.

First, Please explain me the choice of MK-801 concentrations. Is that better to use dose-response curve prior to drug administration?

Second, please explain absense of MK-801 on daily sleep time

THird, I recommend to use molecular methods, e.g qPCR to prove gene expression induced by NMDA antagonists.

Author Response

The manuscript entitled Effects of transient administration of the NMDA receptor antagonist MK-801 in Drosophila melanogaster activity, sleep, and negative geotaxis is presented for the peer review.  In  the study,authors analyzed the effects of MK-801 on the locomotion, sleep, and negative geotaxis of the fruit fly Drosophila melanogaster. Authors used fruit fly as the popular animal model for neuronal studies. You did great job to perform such interesting project.

Good idea to use combined effects of MK-801 on general activity, sleep characteristics, and negative geotaxis.

- We thank the Reviewer for the thoughtful comments.

I have several suggestions to authors.

First, Please explain me the choice of MK-801 concentrations. Is that better to use dose-response curve prior to drug administration?

- We agree with the Reviewer that the reasoning behind the choice of drug concentration is important for the study. In the original manuscript, we have described in the Methods section the following:

 “Drug concentrations were chosen based on a previous publication [38], where the continuous application of ~0.3mM MK-801 was sufficient to induce sleep deficits in flies, while the 0.15mM and 0.6mM concentrations were added to observe possible dose-response effects.”

Nevertheless, to make it clear to the reader, we now have added to the Discussion section (lines 199-207) of the revised version of the manuscript an additional sentence explaining our rationale for concentration choices. It reads:

“In this study, we implemented an NMDAR-inhibition model for sensorimotor deficits by administering the MK-801 compound [27]. Although widely used in rodent research, the effects of this NMDA-inhibition model in adult Drosophila melanogaster have not been characterised, apart from one study demonstrating that continuous administration of MK-801 in the flies’ food reduces sleep duration [38]. Here, we have administered MK-801 for 24 hours or 7 days, which was withdrawn prior to the tests evaluating forced-climbing locomotion (negative geotaxis), general activity, and sleep characteristics. The concentrations chosen (0.15, 0.3, and 0.6 mM) were chosen based on the previous report employing MK-801 in Drosophila (~0.3mM) [38].”

We hope this additional information helps to clarify our reasoning for the choice of MK-801 concentrations.

Second, please explain absense of MK-801 on daily sleep time

We are grateful for the Reviewer’s suggestion to address potential reasons for the discrepancy between our findings and those of previous studies regarding the average sleep time. Thus, we have discussed this point in the new version of our manuscript (lines 222-231), as follows:

“It is worth noting that our findings differ from a previous study that reported reduced sleep time in flies treated with similar doses of MK-801 [38]. We attribute this discrepancy to the different administration protocols used in the two studies, as we removed the treatment before the 3-day assessment of general activity and sleep in our study. This interpretation is supported by data from rodents, which have shown that NMDA antagonists consistently impair electrophysiological sleep features [44,45], but are unable to reduce the sleep-wake times in several instances [46,47]. Therefore, we believe that the transient nature of our drug administration protocol likely hindered the influence of MK-801 on Drosophila sleep time, although it is possible that other sleep features not measured in this study were impaired.”

Third, I recommend to use molecular methods, e.g qPCR to prove gene expression induced by NMDA antagonists.

- We are grateful for the Reviewer's suggestion to explore the potential genetic changes induced by MK-801 in more depth. While we agree that this would be an interesting direction for future research, we believe it falls outside the scope of the current manuscript as our article is intended to be a proof-of-principle study demonstrating that NMDA antagonists can produce similar behavioral effects in flies as those seen in rodent models.

However, we understand the importance of supporting upcoming research on the genetic effects of NMDA receptor antagonists and have included a short statement in the Conclusion section of the revised manuscript to this effect (lines 241-247). We hope that this will encourage other researchers to investigate this topic in more detail in the future. It reads as the following:

“Overall, our findings suggest that Drosophila melanogaster may be a valuable tool for studying schizophrenia-like locomotor phenotypes induced by acute administration of MK-801. Moreover, due to the benefits of the use of Drosophila as a model organism, including the simplicity of the experimental setup and the ability to perform large-scale genetic manipulations, we believe it could be an attractive model for studying the effects of NMDA antagonists on behaviour and for identifying potential therapeutic targets for schizophrenia.”

Reviewer 2 Report

Manuscript presents a pilot study providing proof of principle that the NMDA antagonist MK-801 can be effectively given to Drosophila to produce effects similar to those seen in rodent models.  Work is generally well-performed and well-described. A few small additions to the discussion would increase the impact even further for the reader.

1. The 5 hour length of the climbing assay raises the question whether the effects can be attributed entirely to hyperactivity or whether reduced fatigue in drug-treated animals may play a role in the increased climbing? This could be distinguished by looking at the last hour of climbing separately, which could be extracted from the existing data, I think? If the differences in the last hour are similar to those in the first hour, then this could be ruled out. If not, authors may have discovered another interesting phenotype from the drug that could be added to the discussion.

2. It would be helpful for authors to address in discussion potential reasons why their findings on the effect of chronic administration on sleep turned out differently than previous studies? Although this would be speculative to some extent, it may be helpful for other researchers who might consider using this drug paradigm.

3. Perhaps this may be due to my aging eyes, but I found the color coding a bit difficult to see in the figures, especially figure 1. Might be helpful to some readers to use colors across a broader range, rather than only blues and greens.

Author Response

Manuscript presents a pilot study providing proof of principle that the NMDA antagonist MK-801 can be effectively given to Drosophila to produce effects similar to those seen in rodent models.  Work is generally well-performed and well-described. A few small additions to the discussion would increase the impact even further for the reader.

  1. The 5 hour length of the climbing assay raises the question whether the effects can be attributed entirely to hyperactivity or whether reduced fatigue in drug-treated animals may play a role in the increased climbing? This could be distinguished by looking at the last hour of climbing separately, which could be extracted from the existing data, I think? If the differences in the last hour are similar to those in the first hour, then this could be ruled out. If not, authors may have discovered another interesting phenotype from the drug that could be added to the discussion.

- We are grateful for the Reviewer’s ingenious comment. As shown in the manuscript, we have analysed the climbing activity data using the two-way ANOVA test and Holm-Sidak’s posthoc test to verify differences among treatment groups. However, we inadvertently omitted the p-values for the other variables tested by the two-way ANOVA: time and time-treatment interaction. In the new version of the manuscript, we describe these values, which are non-significant for all groups (see below). These results indicate that time does not affect any group, and neither does MK-801 influences the activity over time. In other words, no group seemed to get significantly fatigued in our protocol, and the drug did not seem to make the flies more tired or energetic over time.

The new description in the Results section (lines 170-179) reads as the following:

“We administered three concentrations of MK-801 (0.15, 0.30, or 0.60 mM) via the food of wild-type adult flies, both acutely (24 hours) and chronically (7 days), followed by a forced climbing assay (Figure 1A). All tested concentrations significantly increased locomotion during the forced climbing assay, where the flies were mechanically tapped down every minute for five hours and the average movement due to innate negative geotaxis was measured (Figure 1B-C) (2-way ANOVA, p<0.0001 and p=0.0002 for 24h- and 7d-treatment effects, respectively. Confirmed by Holm-Sidak’s post hoc tests for each group; results shown in the figure). No effects of time or drug-time interaction were observed, suggesting that flies were not fatigued in our protocol, as there was no significant change of moves over time, and that the treatment did not differentially induce changes over time (ptime=0.946 and 0.865; pinteraction=0.998 and 0.999; for 24h- and 7d-treated flies, respectively).”

  1. It would be helpful for authors to address in discussion potential reasons why their findings on the effect of chronic administration on sleep turned out differently than previous studies? Although this would be speculative to some extent, it may be helpful for other researchers who might consider using this drug paradigm.

We would like to thank the Reviewer for their suggestion to address the differences between our results and those of previous studies regarding average sleep time. We agree that this would be beneficial for other researchers who may be considering using this drug paradigm. We have therefore included a discussion of this point in our revised manuscript (lines 222-231):

“It is worth noting that our findings differ from a previous study that reported reduced sleep time in flies treated with similar doses of MK-801 [38]. We attribute this discrepancy to the different administration protocols used in the two studies, as we removed the treatment before the 3-day assessment of general activity and sleep in our study. This interpretation is supported by data from rodents, which have shown that NMDA antagonists consistently impair electrophysiological sleep features [44,45], but are unable to reduce the sleep-wake times in several instances [46,47]. Therefore, we believe that the transient nature of our drug administration protocol likely hindered the influence of MK-801 on Drosophila sleep time, although it is possible that other sleep features not measured in this study were impaired.”

  1. Perhaps this may be due to my aging eyes, but I found the color coding a bit difficult to see in the figures, especially figure 1. Might be helpful to some readers to use colors across a broader range, rather than only blues and greens.

We thank the Reviewer for the suggestion and thoughtful words. We have revised the figures as suggested in order to improve their accessibility for all readers. We hope that these changes will be helpful and that the revised figures are more visually appealing.